# ISBNet: Instance-aware Selective Branching Networks

## Abstract

Recent years have witnessed growing interests in designing efficient neural networks and neural architecture search (NAS). Although remarkable efficiency and accuracy have been achieved, existing expert designed and NAS models neglect the fact that input instances are of varying complexity and thus different amounts of computation are required. Inference with a fixed model that processes all instances through the same transformations would incur computational resources unnecessarily. Customizing the model capacity in an instance-aware manner is required to alleviate such a problem. In this paper, we propose a novel *Instance-aware Selective Branching Network-ISBNet* to support efficient instance-level inference by selectively bypassing transformation branches of insignificant importance weight. These weights are dynamically determined by a lightweight hypernetwork *SelectionNet* and recalibrated by *gumbel-softmax* for sparse branch selection. Extensive experiments show that *ISBNet* achieves extremely efficient inference in terms of parameter size and FLOPs comparing to existing networks. For example, *ISBNet* takes only 8.70% parameters and 31.01% FLOPs of the efficient network MobileNetV2 with comparable accuracy on CIFAR-10.

## 1 Introduction

Deep convolutional neural networks (CNNs) (He et al., 2016; Zoph et al., 2018) have revolutionized computer vision with increasingly larger and more sophisticated architectures. These model architectures have been designed and calibrated by domain experts with rich engineering experience. To achieve good inference results, these models typically comprise hundreds of layers and contain tens of millions of parameters and consequently consume substantial amounts of computational resources for both training and inference. Recently, there has been a growing interest in efficient network design (Howard et al., 2017; Iandola et al., 2016; Zhang et al., 2018; Sandler et al., 2018) and neural architecture search (NAS) (Zoph et al., 2018; Real et al., 2018; Liu et al., 2018b), respectively with the objective of devising network architectures that are efficient during inference and automating the architecture design process.

Many efficient architectures have indeed been designed in recent years. SqueezeNet (Iandola et al., 2016) and MobileNet (Howard et al., 2017) substantially reduce parameter size and computation cost in terms of FLOPs on mobile devices. More recent works such as MobileNetV2 (Sandler et al., 2018) and ShuffleNetV2 (Ma et al., 2018) further reduce the FLOPs. It is well recognized that devising these architectures is non-trivial and requires engineering expertise.

Automating the architecture design process via neural architecture search (NAS) has attracted increasing attention in recent years. Mainstream NAS algorithms (Zoph & Le, 2016; Zoph et al., 2018; Real et al., 2018) search for the network architecture iteratively. In each iteration, an architecture is proposed by a controller, and then trained and evaluated. The evaluation performance is in turn exploited to update the controller. This process is incredibly slow because both the controller and each derived architecture require training. For instance, the reinforcement learning (RL) based controller NASNet (Zoph et al., 2018) takes 1800 GPU days and the evolution algorithm based controller AmoebaNet (Real et al., 2018) incurs 3150 GPU days to obtain the best architecture. Many acceleration methods (Baker et al., 2017; Liu et al., 2018a; Bender et al., 2018; Pham et al., 2018) have been proposed to accelerate the search process, and more recent works (Liu et al., 2018b;

Xie et al., 2018; Wu et al., 2018; Cai et al., 2018) remove the controller and instead optimize the architecture selection and parameters together with gradient-based optimization algorithms.

While both expert designed and NAS searched models have produced remarkable efficiency and prediction performance, they have neglected one critical issue that would affect inference efficiency. The architectures of these models are fixed during inference time and thus not adaptive to the varying complexity of input instances. However, in real-world applications, there are only a small fraction of input instances requiring deep representations (Wang et al., 2018; Huang et al., 2017a). Consequently, expensive computational resources would be wasted if all instances are treated equally. Designing a model with sufficient representational power to cover the hard instances, and meanwhile a finer-grained control to provide just necessary computation dynamically for instance of varying difficulty is therefore essential.

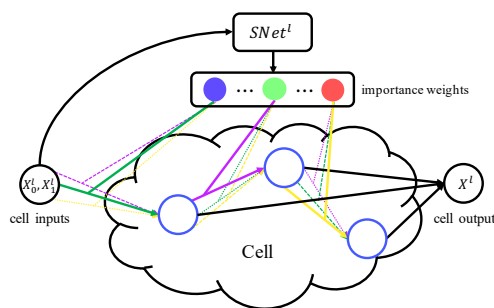

Figure 1: Illustration of the building block *Cell* of *ISBNet*. The *cell* comprises a *SelectionNet* and $N$ interconnected nodes. Each node connection transforms via $B$ diverse branches. The branch selection is guided by importance weights from the *SelectionNet* which is recalibrated with Gumbel-softmax.

In this paper, we propose *ISBNet* to address the aforementioned issue with its building block *Cell* as illustrated in Figure 1. Following the widely adopted strategy in NAS (Zoph et al., 2018; Pham et al., 2018; Liu et al., 2018b; Xie et al., 2018), the backbone network is a stack of $L$ structurally identical cells, receiving inputs from their two previous cells and each cell contains $N$ inter-connected computational *Nodes*. The architecture of *ISBNet* deviates from the conventional wisdom of NAS which painstakingly search for the connection topology and the corresponding transformation operation of each connection. In *ISBNet*, each node is instead simply connected to its prescribed preceding node(s) and each connection transforms via a candidate set of $B$ operations (branches). To allow for instance-aware inference control in the branch level, we integrate $L$ lightweight hypernetworks *SelectionNets*, one for each cell to determine the importance weight of each branch. Gumbel-softmax (Jang et al., 2016; Maddison et al., 2016) is further introduced to recalibrate these weights, which enables efficient gradient-based optimization during training, and more importantly, leads to sparse branch selection during inference for efficiency.

The contributions of *ISBNet* can be summarized as follows:

- *ISBNet* is a general architecture framework combining advantages from both efficient network design and NAS, whose components are readily customizable.

- *ISBNet* is a novel architecture supporting the instance-level selective branching mechanism by introducing lightweight *SelectionNets*, which improves inference efficiency significantly by reducing redundant computation.

- *ISBNet* successfully integrates gumbel-softmax to the branch selection process, which enables direct gradient descent optimization and is more tractable than RL-based method.

- *ISBNet* achieves state-of-the-art inference efficiency in terms of parameter size and FLOPs and inherently supports applications requiring fine-grained instance-level control.

Our experiments show that *ISBNet* is extremely efficient during inference and successfully selects only vital branches on a per-input basis. In particular, with a minor 1.07% accuracy decrease, *ISBNet* reduces the parameter size and FLOPs by 10x and 11.31x respectively comparing to the NAS searched high-performance architecture DARTS (Liu et al., 2018b). Furthermore, with a tiny model of 0.57M parameters, *ISBNet* achieves much better accuracy while with only 8.03% and 30.60% inference time parameter size and FLOPs comparing to the expert-designed efficient network ShuffleNetV2 1.5x (Ma et al., 2018). We also conduct ablation studies and visualize the branch selection process to understand the proposed architecture better. The main results and findings are summarized in Sec 4.2 and Sec 4.3.

## 2    RELATED WORK

**Efficient Network Design.** Designing resource-aware networks (Iandola et al., 2016; Gholami et al., 2018; Ma et al., 2018; Sandler et al., 2018; Gholami et al., 2018) has attracted a great deal of attention in recent years. However, these works mainly focus on reducing parameter size and inference FLOPs in many works. For instances, SqueezeNet (Hu et al., 2018) reduces parameters and computation with the fire module; MobileNetV2 (Sandler et al., 2018) utilize depth-wise and point-wise convolution for more parameter-efficient convolutional neural networks; ShuffleNetV2 (Ma et al., 2018) proposes lightweight group convolution with channel shuffle to facilitate the information flowing across the channels. To make inference efficient, many of these transformations are introduced to the candidate operation set in *ISBNet*.

Many recent works explore conditional (Wang et al., 2018) and resource-constrained prediction (Huang et al., 2017a) for efficiency. SkipNet (Wang et al., 2018) introduces a gating hypernetwork to determine whether to bypass each residual layer (He et al., 2016) conditional on the current input instance. Compared with SkipNet, *ISBNet* provides more efficient and diversified branch selections for the backbone network and the hypernetworks in *ISBNet* are optimized in an end-to-end training manner instead of generally less tractable policy gradient (Williams, 1992). MSDNet (Huang et al., 2017a) supports budgeted prediction within prescribed computational resource constraint during inference by inserting multiple classifiers into a 2D multi-scale version of DenseNet (Huang et al., 2017b). By early-exit into a classifier, MSDNet can provide approximate predictions with minor accuracy decrease. Functionally, *ISBNet* also supports budgeted prediction by dynamically controlling the number of branches selected, therefore per-input inference cost.

**Neural Architecture Search.** Mainstream NAS (Zoph et al., 2018; Real et al., 2018) treats architecture search as a stand-alone process whose optimization is severed from candidate architecture optimization. Search algorithms such as RL-based NAS (Zoph et al., 2018) and evolutionary-based NAS (Real et al., 2018) obtain state-of-the-art architectures at an unprecedented amount of the GPU-time searching cost. Recently, many works have been proposed to accelerate the search pipeline, e.g., via performance prediction (Baker et al., 2017; Liu et al., 2018a), hypernetworks generating initialization weights (Brock et al., 2017), weight sharing (Bender et al., 2018; Pham et al., 2018). These approaches greatly alleviate the search inefficiency while the scalability issue remains unsolved.

A number of proposals (Liu et al., 2018b; Wu et al., 2018; Cai et al., 2018) instead integrate the architecture search process and architecture optimization into the same gradient-based optimization framework. In particular, DARTS (Liu et al., 2018b) relaxes discrete search space to be continuous by introducing operation mixing weights to each connection and optimizes these weights directly with gradient back-propagated from validation loss. Similarly, the discrete search space in SNAS (Xie et al., 2018) is modeled with sets of one-hot random variables for each connection, which is made differentiable by relaxing the discrete distribution with continuous *concrete distribution* (Jang et al., 2016; Maddison et al., 2016). In terms of architecture optimization, *ISBNet* also relaxes the discrete branch selection to continuous importance weights optimized by gradient descent; while instead of direct optimization on the weights, *SelectionNets* are introduced to dynamically generate these weights which are more effective and meanwhile bring about larger model capacity. Further, *SelectionNets* enable instance-level architecture customization rather than finding a fixed model.

## 3    INSTANCE-AWARE SELECTIVE BRANCHING NETWORK

### 3.1    THE BACKBONE NETWORK

The backbone network is constructed with a stack of $L$ cells, each of which is a directed acyclic graph consisting of an ordered sequence of $N$ intermediate nodes. As is illustrated in Figure 1, $x_0^l$ and $x_1^l$ are the cell input nodes from the two preceding cells; each intermediate node $x_i^l (i \geq 2)$ of the $l_{th}$ cell forms a latent representation and receives $n$ input nodes[1] from its preceding nodes:

---

[1] $n$ can be larger than 2 for deeper and wider local representation. E.g., $n = i - 1$ for each $x_i^l$ leads to dense connection, i.e., DenseNet (Huang et al., 2017b).

$$x_i^l = \sum_{j \in \mathbb{S}_i^l} \mathcal{F}_{j,i}(x_j^l), \mathbb{S}_i^l \subset \{0, 1, \cdots, i-1\} \wedge |\mathbb{S}_i^l| = n \tag{1}$$

Thereby, each cell contains $C = n \cdot N$ connections in total. The connection passes information from node $x_j^l$ to $x_i^l$ after the aggregation of a candidate set of $B$ branches of transformation inspired from widely-adopted transformations in NAS (Pham et al., 2018; Liu et al., 2018b; Xie et al., 2018) and efficient network design (Iandola et al., 2016; Sandler et al., 2018; Zhang et al., 2018):

$$\mathcal{F}_{j,i}(x_j^l) = \sum_{b=1}^{B} w_b \cdot \mathcal{F}_b(x_j^l) \tag{2}$$

where $w_b$ here represents the importance of the $b_{th}$ branch (operation) of the connection and is dynamically generated by the cell hypernetwork rather than a fixed learned parameter as is in existing NAS methods (Liu et al., 2018b; Xie et al., 2018). We shall introduce the hypernetwork in Section 3.2. Finally, the output of the cell $x_{out}^l$ is aggregated by *concatenating* the output from all the *intermediate* nodes. We shall use superscript $l$, subscript $c$ and $b$ to index the cell, connection, and branch respectively.

Recent work (Xie et al., 2019) reveals that architectures with randomly generated connection achieve surprisingly competitive results comparing to best NAS models, which is confirmed empirically in our experiments on smaller datasets. In this paper, we thus mainly focus on the branch transformation and selection part and their impact on the inference efficiency instead of specifying a detailed connection topology. Under this architecture formulation framework, we can readily adjust the number of candidate branches $B$ and also the specific transformations before training, customizing model capacity and efficiency respectively depending on the difficulty of the task and resource constraints in deployment.

### 3.2 *SelectionNet* FOR WEIGHT RECALIBRATION

To support instance level inference control, we introduce $L$ lightweight hypernetworks *SelectionNet*, one for each cell. Each *SelectionNet SNet$^l$* receives the same input as the $l_{th}$ cell, specifically the two output nodes $x_{out}^{l-2}, x_{out}^{l-1}$ (i.e. $x_0^l, x_1^l$) from the preceding cells, and concurrently produce $C$ sets of recalibration weights, one for each connection of the cell:

$$\boldsymbol{W}^l = SNet^l(x_0^l, x_1^l) \tag{3}$$

where $\boldsymbol{W}^l \in \mathbb{R}^{C \times B}$ is the recalibration weight matrix for the $l_{th}$ cell. The *SelectionNet SNet$^l$* dynamically generates these weights with a pipeline of $m = 2$ convolutional blocks, a global average pooling and finally an affine transformation. For the $m$ convolutional transformation, we adopt separable convolution (Sandler et al., 2018) which contains a point-wise convolution and a depth-wise convolution of stride 2 and kernel size $5 \times 5$. The stride reduces the parameter size and computation of *SNet$^l$*, and the larger kernel size for depth-wise convolution incurs negligible overhead while extracts features for the immediate weight generation with a larger local receptive field.

The recalibration weights given by the SelectionNet is reminiscent of convolutional attention mechanism (Hu et al., 2018; Woo et al., 2018; Newell et al., 2016), where attention weights are determined dynamically by summarizing information of the immediate input and then exploited to recalibrate the relative importance of different input dimensions, e.g., channels in SENet (Hu et al., 2018). In *ISBNet*, the recalibration weights are introduced to the branch. Particularly, each candidate operation of the connection is coupled with a rescaling weight.

The gumbel-softmax (Jang et al., 2016; Maddison et al., 2016) technique and the reparameterization trick (Kingma & Welling, 2013) is introduced to further recalibrate these weights generated by the SelectionNet, to enable efficient gradient-based optimization for the whole network during training, and more importantly, ensure a sparse selection of important branches during inference. More specifically, each set of importance weights $\boldsymbol{W}_c^l \in \mathbb{R}^B$ for the $c_{th}$ connection of the $l_{th}$ cell ($C_c^l$) after the following recalibration of the gumbel-softmax follows *concrete distribution* (Maddison et al., 2016) controlled by a temperature parameter $\tau$:

$$\widetilde{w}_{c,b}^{l} = \frac{\exp((w_{c,b}^{l} + G_{c,b}^{l})/\tau)}{\sum_{b'=1}^{B} \exp((w_{c,b'}^{l} + G_{c,b'}^{l})/\tau)}, \tau > 0 \tag{4}$$

where $\widetilde{w}_{c,b}^{l}$ is then directly used for branch recalibration as in Equation 2, and $G_{c,b}^{l} = -\log(-\log(U_{c,b}^{l}))$ here is a *gumbel* random variable coupling with $b_{th}$ branch by sampling $U_{c,b}^{l}$ from $Uniform(0,1)$ (Jang et al., 2016). The *concrete distribution* (Maddison et al., 2016) suggests that (1) $\widetilde{w}_{c,b} = \frac{1}{B}$, as $\tau \to +\infty$, and more importantly (2):

$$p(\lim_{\tau \to 0} \widetilde{w}_{c,b}^{l} = 1) = \exp(w_{c,b}^{l}) / \sum_{b'=1}^{B} \exp(w_{c,b'}^{l}) \tag{5}$$

Therefore, high temperature leads to uniform dense branch selection while lower temperature tends to sparsely sample branches following a *categorical distribution* parameterized by $softmax(\boldsymbol{W}_{c}^{l})$.

### 3.3 OPTIMIZATION AND INFERENCE FOR *ISBNet*

With the continuous relaxation of the gumbel-softmax (Jang et al., 2016; Maddison et al., 2016) and the reparameterization (Kingma & Welling, 2013), the branch selection process of the SelectionNets is made directly differentiable with respect to the weight $w_{c,b}^{l}$. In particular, the gradient $\frac{\partial \mathcal{L}}{\partial \widetilde{w}_{c,b}^{l}}$ backpropagated from the loss function $\mathcal{L}$ to $\widetilde{w}_{c,b}^{l}$ through the backbone network can be directly backpropagated to $w_{c,b}^{l}$ with low variance (Maddison et al., 2016), and further to the $l_{th}$ SelectionNet unimpededly. Therefore, parameters of the whole network can be optimized in an end-to-end manner by gradient descent.

The temperature $\tau$ of Equation 4 regulates the sparsity of the branch selection. A relatively higher temperature forces the weights to distribute more uniformly so that all the branches of each connection are efficiently trained. While a low temperature instead tends to sparsely sample one branch from the categorical distribution parameterized by the importance weights dynamically determined by *SelectionNets*, thus supporting finer-grained instance-level inference control by bypassing unimportant branches. To leverage both characteristics, we propose a *two stage training scheme* for *ISBNet*: (1) the first stage pretrains the whole network with a *fixed* relatively high temperature till convergence. (2) the second stage fine-tunes the parameters with $\tau$ steadily annealing to a relatively low temperature. The first stage ensures that branches are sufficiently optimized before the instance-aware selection and the fine-tuning in the second stage helps maintain the performance of *ISBNet* under sparse branch selection during inference.

To further promote inference efficiency and reduce redundancy, a regularization term is explicitly introduced in the fine-tuning stage which takes into account the *expectation* of the resource consumption $\mathcal{R}$ in the final loss function $\mathcal{L}$ for *correctly classified* instances:

$$\mathcal{L} = \mathcal{L}_{CE} + \lambda_1 ||w||_2^2 + \lambda_2 \mathbf{1}_{\hat{\mathbf{y}}=\mathbf{y}} \log \mathbb{E}[\mathcal{R}]$$
$$\approx \mathcal{L}_{CE} + \lambda_1 ||w||_2^2 + \lambda_2 \mathbf{1}_{\hat{\mathbf{y}}=\mathbf{y}} \log \sum_{l=1}^{L} \sum_{c=1}^{C} \sum_{b=1}^{B} \widetilde{w}_{c,b}^{l} \cdot \mathcal{R}(\mathcal{F}_{c,b}^{l}(\cdot)) \tag{6}$$

where $\mathcal{L}_{CE}$ and $\lambda_1 ||w||_2^2$ denotes the cross-entropy loss and the weight decay term, $\mathbf{y}$ is the ground truth class label, $\hat{\mathbf{y}}$ the prediction, $\lambda_2$ controls the regularization strength and $\mathcal{R}(\cdot)$ calculates the resource consumption of each operation $\mathcal{F}_{c,o}^{l}(\cdot)$. The operation importance weight $\widetilde{w}_{c,b}^{l}$ represents the probability of the corresponding branch $\mathcal{F}_{c,b}^{l}$ being selected during inference, and therefore the regularization term $\mathbb{E}[\mathcal{R}]$ corresponds to the expectation of the aggregated resource required for each input instance.

The resource regularizer is readily adjustable depending on deployment constraints, which may include the parameter size, FLOPs, and memory access cost (MAC). In this work, we mainly focus on the inference time, specifically FLOPs, which can be calculated beforehand for each branch.

$\mathcal{R}(\mathcal{F}^l_{c,b}(\cdot))$ is thus a constant here, which means that the regularizer $\mathcal{R}$ is also directly differentiable with respect to $\widetilde{w}^l_{c,b}$. We denote *ISBNet* trained with regularization strength $\lambda_2$ as *ISBNet-R-$\lambda_2$*.

During inference, the instance-level selective branching is achieved for each connection $C^l_c$ by selecting branches of the top $k$ largest recalibration weights whose aggregated weight $s^l_c$ just exceeds a threshold $T$. Denoting $\widetilde{\boldsymbol{W}}^l_c$ sorted in descending order as $\widehat{\boldsymbol{W}}^l_c$, then:

$$s^l_c = \min\{s_k : s_k = \sum_{b=1}^{k} \widehat{w}^l_{c,b} \wedge s_k \geq T\} \tag{7}$$

After the selection, the recalibration weight $\widetilde{w}^l_{c,b}$ of the selected branch is rescaled by $\frac{1}{s^l_c}$ to stabilize the scale of the representation. Consequently, the SelectionNet will select only necessary branches[2] for each instance depending on the input difficulty and meanwhile the FLOPs of each branch, i.e., trading off between $\mathcal{L}_{CE}$ and $\mathcal{R}$ in Equation 6. Furthermore, the resource consumption of each instance can be *precisely regulated* in a finer-grained manner by scheduling the threshold dynamically for each connection. In this paper, the same threshold is shared among all connections for simplicity and *ISBNet* inference with the threshold $t$ is denoted as *ISBNet-T-t*.

Under such an inference scheme, the backbone network comprises up to $(2^B-1)^{L \cdot C}$ possible candidate subnets, corresponding to each unique branch selection of all $L \cdot C$ connections. For a small *ISBNet* of 10 cells, with 5 candidate operations, 8 connections per cell, there are $(2^5-1)^{8 \cdot 10} \approx 2 \cdot 10^{119}$ possible candidate architectures of different branch combination, which is orders of magnitudes larger than the search space of conventional NAS (Pham et al., 2018; Liu et al., 2018b; Xie et al., 2018; Cai et al., 2018; Stamoulis et al., 2019).

## 4 EXPERIMENTS

We now compare the performance of *ISBNet* with the best-performing expert-designed efficient networks and NAS architectures using benchmark dataset CIFAR-10 and ImageNet. The experimental details are presented in Sec. 4.1; main results are reported in Sec. 4.2, followed by the visualizations of the branch selection process of *ISBNet* in Sec. 4.3.

### 4.1 EXPERIMENTAL SETUP

**Dataset**   CIFAR-10 contains 50,000 training images and 10,000 test images of $32 \times 32$ pixels in 10 classes. We adopt standard data pre-processing and augmentation pipeline (Liu et al., 2018b; Xie et al., 2018) and apply AutoAugment (Cubuk et al., 2018), cutout (DeVries & Taylor, 2017) of length 16. ImageNet contains 1.2 million training and 50,000 validation images in 1000 classes. We adopt standard augmentation scheme following (Liu et al., 2018b; Xie et al., 2018) and apply label smoothing of 0.1 and AutoAugment. Results are reported with $224 \times 224$ center crop.

**Candidate Operation Set**   The following 5 candidate operations ($B = 5$) are adopted for demonstration and can be readily adjusted in deployment:
- $3 \times 3$ max-pooling
- $3 \times 3$ avg-pooling
- skip connection
- $3 \times 3$ separable-conv
- $5 \times 5$ separable-conv

In particular, separable-conv stands for two separable convolution (Howard et al., 2017) of *ReLU-Conv-Conv-BN*. Skip connection allows for efficient representation forwarding; pooling layers are computationally lightweight with no parameter; and separable-conv dominates the parameter size and computation in each connection. The three types of operations support *trade-off* between representation power and efficiency for the branch selection of each connection.

**Temperature Annealing Scheme**   In the pre-training stage, the temperature $\tau$ is fixed to 3 till convergence. In the fine-tuning stage, $\tau$ is initialized to 1.0 and is further annealed by $\exp(-0.0006) \approx 0.999$ every epoch to 0.5 for CIFAR-10. The temperature is 3 throughout for ImageNet.

---

[2]At least one branch will be selected for each connection.

**Architecture Details**    For CIFAR-10, we evaluate two *ISBNet* architectures of different size for demonstration: (1) *ISBNet(S)*, a small network with $L = 5$ cells and 15 initial channels; (2) *ISBNet(M)*, a medium network with $L = 10$ cells and 20 initial channels. For ImageNet, we evaluate a medium network with $L = 10$ cells and 32 initial channels.

All the architectures contain $N = 4$ intermediate nodes in each cell and we adopt a simple node connection strategy of connecting to the two preceding nodes (i.e. $x_{i-1}^l$ and $x_{i-2}^l$) for CIFAR-10, and connecting to the preceding node ($x_{i-1}^l$) and randomly $x_0^l$ or $x_1^l$ for ImageNet. Further, nodes directly connected to the input nodes are downsampled with stride 2 for $\frac{L}{3}$-th and $\frac{2L}{3}$-th cells. An auxiliary classifier with weight 0.4 is connected to the output of $\frac{2L}{3}$-th cell for extra regularization.

**Optimization Details**    For CIFAR-10, we apply SGD with momentum 0.9 and weight decay $3 \cdot 10^{-4}$ for 1200 epochs for both training stages. The learning rate is initialized to 0.025 and 0.005 for the pre-training and fine-tuning stage respectively. We use batch size 256/128 for CIFAR-10 (*ISBNet(S)/ISBNet(M)*) and batch size 128 for ImageNet to fit the whole network into one Titan RTX. For ImageNet, we apply SGD with nesterov-momentum 0.9 and weight decay $3 \times 10^{-5}$ for 250 epochs. We adopt drop-connection and drop-branch linearly to 0.1 and 0.7/0.5 respectively for CIFAR-10/ImageNet. The learning rate is annealed to zero with one cycle of cosine learning rate scheduler (Loshchilov & Hutter, 2016).

## 4.2    *ISBNet* PERFORMANCE EVALUATION

| Architecture | Test Error (%) | Training / Inference Params (M) | Training / inference FLOPs (M) | Search Method | Search Space | Search Cost (GPU days) |
|---|---|---|---|---|---|---|
| ResNet101 (He et al., 2016) | 6.25 | 42.51 | 2520 | manual | − | - |
| DenseNet-BC (Huang et al., 2017b) | 3.46 | 25.6 | 9345 | manual | − | - |
| MobileNetV2 1.0× (Sandler et al., 2018) | 5.56 | 2.30 | 94.42 | manual | − | - |
| ShuffleNetV2 1.5× (Ma et al., 2018) | 6.36 | 2.49 | 95.70 | manual | − | - |
| NASNet-A (Zoph et al., 2018) | 2.65 | 3.3 | − | RL | cell | 1800 |
| AmoebaNet-A (Real et al., 2018) | 3.34 | 3.2 | − | evolution | cell | 3150 |
| ENAS (Pham et al., 2018) | 2.89 | 4.6 | 86.86 | RL | cell | 2.2 |
| DARTS (Liu et al., 2018b) | 3.00 | 3.3 | 542.0 | gradient | cell | 1.9 |
| SNAS (Xie et al., 2018) | 3.10 | 2.3 | − | gradient | cell | 3 |
| *ISBNet(S)-w/o-SelectionNet* | 4.78 | 0.46 | 77.34 | gradient | layer-wise | - |
| *ISBNet(S)-Softmax-T*-0.8 | 4.37 / 5.27 | 0.57 / 0.55 | 84.65 / 80.99 | gradient | layer-wise | - |
| *ISBNet(S)-Softmax-T*-0.6 | 4.37 / 15.2 | 0.57 / 0.48 | 84.65 / 72.29 | gradient | layer-wise | - |
| *ISBNet(S)-R*-0.0-*T*-0.8 | 3.66 / 4.07 | 0.57 / 0.33 | 84.65 / 47.91 | gradient | layer-wise | - |
| *ISBNet(S)-R*-0.0-*T*-0.6 | 3.66 / 4.26 | 0.57 / 0.31 | 84.65 / 45.61 | gradient | layer-wise | - |
| *ISBNet(S)-R*-0.1-*T*-0.8 | 3.66 / 4.21 | 0.57 / 0.31 | 84.65 / 43.82 | gradient | layer-wise | - |
| *ISBNet(S)-R*-0.5-*T*-0.8 | 3.66 / 5.85 | 0.57 / 0.20 | 84.65 / 29.28 | gradient | layer-wise | - |
| *ISBNet(M)-R*-0.0-*T*-0.8 | 2.84 / 3.26 | 1.86 / 1.02 | 267.3 / 139.5 | gradient | layer-wise | - |
| *ISBNet(M)-R*-0.1-*T*-0.8 | 2.84 / 3.44 | 1.86 / 0.89 | 267.3 / 119.6 | gradient | layer-wise | - |
| *ISBNet(M)-R*-0.5-*T*-0.8 | 2.84 / 4.50 | 1.86 / 0.66 | 267.3 / 74.90 | gradient | layer-wise | - |

Table 1: Comparison of *ISBNet* with other state-of-the-art architectures on CIFAR-10 dataset. Results of *ISBNet* are reported in *full model* / *selective branching* respectively.

**Overall Results and Discussion.**    Table 1 summarizes the overall performance on CIFAR-10 of *ISBNet* under different inference threshold $T$ and resource constraint strength $R$. In terms of training efficiency, *ISBNet* takes only 2.5 and 5.5 GPU training days for *ISBNet(S)* and *ISBNet(M)* respectively without any architecture searching, which is up to three orders of magnitudes less time than conventional evolution-based NAS or RL-based NAS thanks to the efficient network design and the end-to-end gradient-based optimization.

As for inference time performance, *ISBNet* reduces a drastic amount of the parameter size and FLOPs comparing to baseline networks. Specifically, with comparable accuracy, *ISBNet(S)-R*-0.5-*T*-0.8 only takes 0.20M parameters and 29.28M FLOPs on average during inference, which is only 8.03% and 30.60% of efficient network ShuffleNetV2 1.5×; *ISBNet(S)-R*-0.0-*T*-0.8 achieves up to 10x and 11x parameter size and FLOPs reduction than DARTS with 1.07% accuracy decrease. The drastic parameter size and FLOPs reduction demonstrate that the selective branching mechanism in *ISBNet* enables extremely efficient instance-level prediction. This is also corroborated by the significant reduction of the parameter size and FLOPs from training to inference of *ISBNet*, i.e. from 0.57M parameters and 84.65M FLOPs to 0.33M and 47.91M in *ISBNet(S)-R*-0.0-*T*-0.8.

| Architecture | Test Error (%) | Params (M) | FLOPs (M) | Search Method | Search Space | Search Cost (GPU days) |
|---|---|---|---|---|---|---|
| ShuffleNet 2× (Zhang et al., 2018) | 29.1 | 5 | 524 | - | | |
| 1.0 MobileNet (Howard et al., 2017) | 29.4 | 4.2 | 569 | manual | - | - |
| 0.75 MobileNet (Howard et al., 2017) | 31.6 | 2.6 | 325 | manual | - | - |
| NASNet0-A (Zoph et al., 2018) | 26.0 | 5.3 | 564 | RL | cell | 1800 |
| AmoebaNet-A (Real et al., 2018) | 25.5 | 5.1 | 555 | evolution | cell | 3150 |
| DARTS Liu et al. (2018b) | 26.9 | 4.9 | 595 | gradient | cell | 4 |
| *ISBNet-R*-0.0-*T*-1.0 | 29.9 | 4.9 | 485 | gradient | layer-wise | - |
| *ISBNet-R*-0.0-*T*-0.9 | 30.1 | 4.0±0.20 | 397±12.6 | gradient | layer-wise | - |
| *ISBNet-R*-0.0-*T*-0.8 | 30.8 | 3.7±0.16 | 360±11.5 | gradient | layer-wise | - |
| *ISBNet-R*-0.0-*T*-0.7 | 32.1 | 3.4±0.13 | 329±10.2 | gradient | layer-wise | - |

Table 2: Statistics and performance of *ISBNet* compared with other architectures on ImageNet dataset.

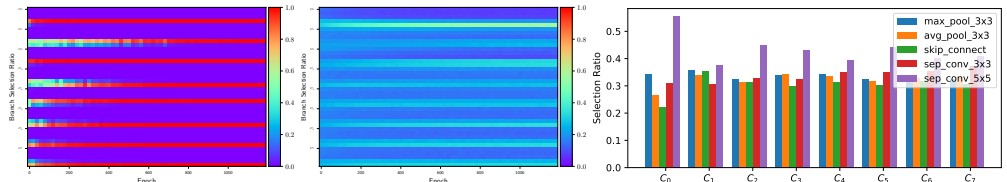

Figure 2: Average Recalibration Weight of the first/last cell during the temperature annealing stage, and inference time Branch Selection Ratio (last cell) of the 5 branches for each connection of *ISBNet(S)-R*-0.0-*T*-0.8.

Results in Table 1 also validate that the resource regularizer effectively regularizes the network for more efficient inference on both parameter size and FLOPs, although only FLOPs is explicitly regularized. Specifically, the larger the regularization strength $\lambda$, the more efficient *ISBNet* is, while at the cost of a minor accuracy decrease. For instance, the inference parameter size of *ISBNet(M)-T*-0.8 is reduced from 1.02M, 0.89M to 0.66M, and FLOPs from 139.46M, 119.56M to 74.90M for regularization strength from 0.0, 0.1 to 0.5 respectively.

The results show that a small *ISBNet* is able to achieve competitive accuracy comparable to the best efficient and NAS-searched models, meanwhile with far fewer inference parameters and FLOPs. This raises questions about the necessity of current laborious architecture search of NAS (Zoph et al., 2018; Real et al., 2018). In this paper, we propose a selective branching mechanism evocative of convolutional attention (Hu et al., 2018; Woo et al., 2018) via the introduction of the hypernetworks *SelectionNet*, which leads to larger model capacity and enables selective branching. With 19.30% more parameters and 8.54% more FLOPs, *ISBNet(S)* integrated with *SelectionNets* achieves 0.41% noticeably higher accuracy. Further trained with gumbel-softmax, *SelectionNets* enables the network to efficiently select necessary branches and customize its architecture on a per-input basis during inference. Gumbel-softmax is necessary for maintaining accuracy because with plain softmax trained *SelectionNets*, *ISBNet(S)-Softmax* suffers from a catastrophic accuracy decrease, from 4.37% to 15.22%, while with limited parameter size and FLOPs reduction with an inference threshold 0.6.

**Accuracy-FLOPs Trade-off.** Table 2 summarizes the performance of *ISBNet* under different thresholds on ImageNet. The results demonstrate that *ISBNet* achieves quite competitive results compared with expert-designed efficient networks and NAS-searched models even with a simple connection scheme and a preconfigured candidate operation set. Further, with the selective branching of SelectionNet, one single network of *ISBNet* supports efficiency-accuracy trade-offs by simply controlling the importance threshold. In particular, *ISBNet* reduce the parameter size by 18.37% and FLOPs by 18.14% with a threshold 0.9 with a minor 0.2 accuracy decrease. With a threshold 0.7, *ISBNet* achieves another 12.24% and 14.02% redundancy reduction respectively. This confirms that networks can support efficient instance-aware inference with the selective branching mechanism.

### 4.3 VISUALIZATION OF SELECTIVE BRANCHING

**Ratio of Selective Branching.** We visualize in Figure 2 the average recalibration weight and branch selection ratio of representative cells in *ISBNet(S)-R*-0.0-*T*-0.8, which shows the ratio during training and in the final model of each branch being selected during inference respectively. An obvious stratified pattern can be observed that one separable-convolution branch gradually dominates the connection in the first cell while in subsequent cells, the branch selection tends to be more

uniform and diversified. This pattern demonstrates that features extracted in lower layers share similar branch transformation where branch pruning can be performed to reduce the parameter size; while the instance-aware efficient inference requires diversified branch selections ascending the layers. Further experiments show that the average number of branches selected in the last cell is 1.1, indicating that only a small number of branches are required for the inference of most instances.

**Qualitative Difference between Instances.**
Denoting instances that the network is confident with in prediction as *easy* instance and uncertain as *hard* instance, we visualize the clustering of *easy* and *hard* instances in Figure 3 to help understand the selective branching mechanism. We find that the certainty of the prediction made by *ISBNet* depends mainly on the image quality. In general, easy instances are more salient (clear with high contrast) while hard instances are more inconspicuous (dark with low contrast). We also compute the accuracy and average FLOPs of each cluster. On average, easy instances achieve much higher classification accuracy with 11.2% fewer FLOPs compared with hard instances. This shows that computation could be greatly reduced without sacrificing accuracy by selective bypassing unimportant branches for relatively easy instances.

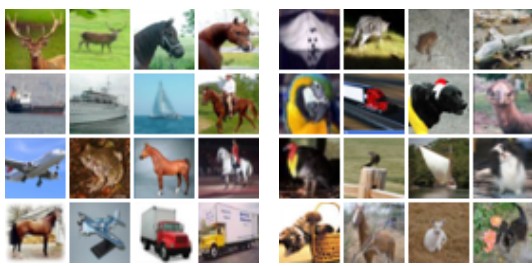

(a) *easy* instances        (b) *hard* instances

Figure 3: Visualization of *easy* and *hard* instances of model *ISBNet(M)-R*-0.0-*T*-0.8 on CIFAR-10. Easy instances are clearer and brighter in general while hard instances are darker and blurry.

## 5 CONCLUSION

In this paper, we have proposed *ISBNet*, a novel network framework with the advantages of both efficient network design and neural architecture search. To achieve efficient instance-aware inference, a series of lightweight hypernetworks are introduced to cells of the backbone network to determine importance weights for selective branching. We have also integrated gumbel-softmax and the reparameterization trick to the branch selection process, which enables accessible and tractable gradient-based end-to-end training, and more importantly, extremely efficient inference. The inference efficiency is further enhanced with the resource-aware regularization. Extensive experiments and visualizations have been conducted, and the results validate the efficiency of instance-aware selective branching inference.

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
