# OpenReview forum: "ISBNet: Instance-aware Selective Branching Networks"
_ICLR.cc/2020/Conference — Reject_

### Official Review · AnonReviewer1 · 2019-10-22
**Official Blind Review #1**

**Rating:** 3

**Review:**

Neural architecture search usually aims to find a single fixed architecture for the task of interest. The paper proposes to condition the architecture on the input instances by introducing a "selection network" that learns to retain a subset of branches in the architecture during each inference pass. The intuition is that easier instances require less compute (hence a shallower/sparser architecture) as compared to the more difficult ones. The authors show improved results on CIFAR-10 and ImageNet in terms of accuracy-latency trade-off over some handcrafted architectures and NAS baselines. The method resembles sparsely gated mixture of experts [1] at a high-level, but has been implemented in a way that better fits the context of architecture search (which is still technically interesting).

My major concern is about the comparisons made against existing works:

The authors argue that instance-aware architecture selection is beneficial. However, there seems to be a misalignment between such a claim and the empirical evidences. Apart from the authors' own controlled experiments (ISBNet w or w/o selection network), in Table 1 & 2 the authors are comparing the accuracy-latency tradeoff of their method (which is resource-aware) against either (1) handcrafted architectures in the literature, or (2) a subset of NAS methods that are *not resource-aware at all*. State-of-the-art resource-aware NAS baselines (that are not instance-aware) such as MNASNet [2] (which is not currently cited), ProxylessNAS, FBNets and EfficientNets are completely missing and are left out from both tables for some reason, which are actually the right baselines to compare against in the reviewer's opinion.

[1] Shazeer, Noam, et al. "Outrageously large neural networks: The sparsely-gated mixture-of-experts layer." arXiv preprint arXiv:1701.06538 (2017).
[2] Tan, Mingxing, et al. "Mnasnet: Platform-aware neural architecture search for mobile." Proceedings of the IEEE Conference on Computer Vision and Pattern Recognition. 2019.


**Experience Assessment:**

I have published one or two papers in this area.

**Review Assessment: Checking Correctness Of Derivations And Theory:**

I carefully checked the derivations and theory.

**Review Assessment: Checking Correctness Of Experiments:**

I carefully checked the experiments.

**Review Assessment: Thoroughness In Paper Reading:**

I read the paper thoroughly.

---

> ### Author Response · Authors · 2019-11-08
> **Response for Reviewer #1**
>
> We would like to thank you for your acknowledgment of our contributions. In this work, we have proposed an instance-aware branch selection framework that can reduce unnecessary computation significantly over the configured backbone network, instead of specifying one network that achieves the best accuracy or fastest inference. We have validated the effectiveness of the proposed framework on dynamically reducing unnecessary parameter size and computation on a per-input basis accordingly. We choose to compare ISBNet with architectures adopted in our experiments mainly due to the candidate operation set adopted. More discussions and justifications are provided in our response to Reviewer #2, please refer to the comment above for details.

---

### Official Review · AnonReviewer2 · 2019-11-05
**Official Blind Review #2**

**Rating:** 3

**Review:**

This paper proposes an instance-aware dynamic network, ISBNet, for efficient image classification. The network consists of layers of cell structures with multiple branches within. During the inference, the network uses SelectionNet to compute a "calibration weight matrix", which essentially controls which branches within the cell should be used to compute the output. Similar to previous works in NAS, this paper uses Gumbel Softmax to compute the branch selection probability. The network is trained to minimize a loss function that considers both the accuracy and the inference cost. Training of the network is divided into two stages: First, a high temperature is used to ensure all the branches are sufficiently optimized, and at the second stage, the authors aneal the temperature. During the inference, branches are selected if their probability computed by Gumbel Softmax is larger than a certain threshold.

Overall, the idea of the paper is clearly presented. The methods used in this paper are similar to previous works on neural architecture search (NAS), but this paper can be seen as a meaningful extension to NAS.

My main concern for this paper is the experiment section.

1) The paper claims that "ISBNet takes only 8.70% parameters and 31.01% FLOPs of the efficient network MobileNetV2 with comparable accuracy on CIFAR-10". Mainstream efficient networks, such as MobileNet and ShuffleNet, are not designed for CIFAR-10 datasets, but ImageNet datasets. Their downsampling strategy is very aggressive, which leads to relatively poor accuracy on CIFAR-10 datasets with 32x32 images. Therefore, it is not fair to compare the MobileNetV2 on CIFAR-10 and claim superiority over it. Compared with other networks customized for the CIFAR-10 dataset, such as NASNet-A, DARTS, and so on, the error rate of ISBNet is significantly worse (>1% higher than DARTS, or >33% relative increase in error rate).

2) In table 2, the paper compares ISBNet's performance on the ImageNet dataset with other baselines. However, the baseline models are not up to date. For example, MobileNetV2 achieves a 28% error rate with 300M FLOPs, FBNet achieves a 27% error rate with 249M FLOPs. These results, however, are not cited in this paper. Particularly, MobileNetV2 is compared against on the CIFAR-10 dataset, but not the ImageNet dataset.

3) ISBNet is not the first instance-aware dynamic network. Previous works such as SkipNet, Soft-Conditional computing [1] explored similar ideas. However, in this paper, there is no comparison with previous dynamic networks.

Overall, I would expect a much stronger experiment section for the paper to be published.

[1] https://arxiv.org/abs/1904.04971

**Experience Assessment:**

I have published in this field for several years.

**Review Assessment: Checking Correctness Of Derivations And Theory:**

I assessed the sensibility of the derivations and theory.

**Review Assessment: Checking Correctness Of Experiments:**

I assessed the sensibility of the experiments.

**Review Assessment: Thoroughness In Paper Reading:**

I read the paper at least twice and used my best judgement in assessing the paper.

---

> ### Author Response · Authors · 2019-11-08
> **Response for Reviewer #2**
>
> First, we thank you for your acknowledgment of our contributions. As for the concerns over the experiment section, we would like to recap that in this work, we have proposed a GENERAL and CUSTOMIZABLE framework for neural networks to inference without UNNECESSARY computation of the BACKBONE network on a PER-INPUT basis. ISBNet can be understood as a general mechanism to effectively and dynamically trade off accuracy and efficiency, instead of one specific state-of-the-art backbone network that achieves the best accuracy or fastest inference. Therefore in the experiment, we mainly focus on validating that ISBNet can support more efficient instance-aware inference with selective branching over the network CONFIGURED with the adopted Candidate Operation Set.
>
> One very important detail that should be taken notice of is that in Sec 4.1, we adopt 5 candidate operations for the Candidate Operation Set. Further explanations over why we adopt these operations should help address concerns over the experiments. Specifically, the 5 operations are the most widely-adopted operations in the search space of NAS models, e.g., NASNet, ENAS, DARTS, SNAS, PNAS [1] and so on. Further, the two separable convolution operations are the workhorse transformation and correspond to the building block of MobileNet instead of OTHER networks.
> Therefore, we argue that:
>
> 1) We have included as many results of different but comparable architectures of both traditional expert-designed and NAS networks as possible (not limited to MobileNetV2), and also ablation study of ISBNet (w/o SelectionNet, w/ Softmax etc.) in Table 1, to support our claims above. The results are mainly reported on CIFAR-10 because the majority of NAS models are searched and benchmarked on CIFAR-10. In terms of accuracy, compared with better-performing networks customized for the CIFAR-10, e.g., DARTS, the error rate of ISBNet(M)-R-0.0-T-1.0 is noticeably better without any architecture search (0.16% higher than DARTS, and the parameter size and FLOPs are 56.36% and 49.32% respectively of DARTS). In terms of accuracy-efficiency trade-offs, the parameter size and FLOPs reduction are more encouraging. More results and discussions are provided in Sec 4.2.
>
> 2) Following conventions of NAS, ISBNet adopts candidate operations that are the building block of MobileNet. This is the main reason that we show results of MobileNet instead of other architectures on ImageNet in Table 2. We support our claims with the results on the larger dataset ImageNet that ISBNet effectively reduces a significant amount of parameter size and FLOPs, and achieves accuracy-FLOPs trade-offs with one single model without any search or engineering work.
>
> Further, it should be noticed that MobileNetV2 and FBNet are inherently more efficient than MobileNet because they adopt significantly different building blocks, i.e. the inverted residual block for MobileNetV2, and for FBNet, candidate blocks of different expansion ratio (similar to the inverted residual) and group convolution for the pointwise convolution with channel shuffle as proposed in ShuffleNet. These blocks are highly optimized over the separable convolution, which achieve significantly better results than MobileNet. To avoid confusion and deviation from the main focus of this work, we do not show the results of these networks. However, we note that ISBNet is a general and flexible framework and thus can readily take in these building blocks for more efficient inference.
>
> 3) We have introduced and discussed the difference between ISBNet and other related works in Sec 2. Compared with SkipNet, ISBNet provides more efficient and diversified branch selections for the backbone network and the hypernetworks in ISBNet are optimized in an end-to-end training manner instead of generally less tractable policy gradient. Further, the backbone of SkipNet is ResNet, which is far less efficient compared with MobileNet. For example, SkipNet-38-Hard only achieves 90.83% accuracy with 58 MFLOPs, and similarly, SkipNet-74-Hard 92.38% with 92 MFLOPs (Table 3 of SkipNet [2]), which is far worse than ISBNet, e.g. ISBNet(S)-R-0.5-T-0.8 achieves 94.15% accuracy with only 29.28 MFLOPs on CIFAR-10. We do not compare with SkipNet in the experiment mainly because these two networks adopt different backbones for the same reason above. As for the contemporary work of Soft-Conditional Computing (CondConv) that we have not yet included in the related work. We note that the idea of CondConv is similar to ours, which also adopts hypernetworks for conditional computation. However, different from the selective branching of ISBNet to reduce the unnecessary computation of each connection, CondConv is designed to be a mixture of experts (i.e. different convolutional branches), which is more like a branch-level weight recalibration as proposed in SENet. We shall include these related works if it may deem fit.
>
> [1] https://arxiv.org/abs/1712.00559
> [2] https://arxiv.org/abs/1711.09485

---

### Decision · Program_Chairs · 2019-12-19

**Decision:**

Reject

**Comment:**

This paper proposes a method for finding neural architecture which, through the use of selective branching, can avoid processing portions of the network on a per-data-point basis.

While the reviewers felt that the idea proposed was technically interesting and well-presented, they had substantial concerns about the evaluation that persisted post-rebuttal, and lead to a consensus rejection recommendation.